# Large Pelvic Mass in a Female Adolescent: Atypical Presentation and Successful Treatment of Extraskeletal Ewing Sarcoma

**DOI:** 10.3390/healthcare11101373

**Published:** 2023-05-10

**Authors:** Federica Perelli, Giuseppe Vizzielli, Anna Franca Cavaliere, Stefano Restaino, Giovanni Scambia, Gian Franco Zannoni, Damiano Arciuolo, Valerio Gallotta

**Affiliations:** 1Obstetrics and Gynecology Unit, Ospedale Santa Maria Annunziata, USL Toscana Centro, Bagno a Ripoli, 50012 Firenze, Italy; 2Clinic of Obstetrics and Gynecology, S. Maria della Misericordia University Hospital, Azienda Sanitaria Universitaria Friuli Centrale (ASUFC), 33100 Udine, Italy; giuseppe.vizzielli@asufc.sanita.fvg.it (G.V.); stefano.restaino@asufc.sanita.fvg.it (S.R.); 3Medical Area Department (DAME), University of Udine, 33100 Udine, Italy; 4Obstetrics and Gynaecology Unit, Dipartimento Centro di Eccellenza Donna e Bambino Nascente, Fatebenefratelli Gemelli Isola Tiberina, 00168 Rome, Italy; annafranca.cavaliere@fbf-isola.it; 5Dipartimento per le Scienze Della Salute Della Donna, del Bambino e di Sanità Pubblica, UOC Ginecologia Oncologica, Fondazione Policlinico Universitario Agostino Gemelli IRCCS, 00168 Rome, Italy; giovanni.scambia@policlinicogemelli.it (G.S.); valerio.gallotta@policlinicogemelli.it (V.G.); 6Pathology Unit, Department of Woman and Child’s Health and Public Health Sciences, Fondazione Policlinico Universitario Agostino Gemelli IRCCS, 00168 Rome, Italy; gianfranco.zannoni@unicatt.it (G.F.Z.); damiano.arciuolo@policlinicogemelli.it (D.A.); 7Pathology Institute, Catholic University of Sacred Heart, 00168 Rome, Italy

**Keywords:** gynecology, gynecologic surgery, female healthcare, Ewing sarcoma, pediatric gynecology, fertility sparing

## Abstract

Extraskeletal Ewing sarcoma is a rare soft tissue tumor primarily affecting pediatric patients. The treatment is currently based on a multidisciplinary approach which allows, in cases of localized disease, good survival rates. We report the case of a 15-year-old female patient with a rapidly growing suspected pelvic mass misdiagnosed following the preliminary radiological exams, which assessed the findings as a mass of ovarian origin. The girl underwent surgery and, thanks to histopathological, immunohistochemical and real-time polymerase chain reaction (RT-PCR) examinations, it was possible to make the right diagnosis and to administer the best treatment in terms of surgery, chemotherapy and radiotherapy, obtaining a long disease-free interval and no recurrence to date.

## 1. Introduction

Extraskeletal Ewing sarcoma (EES) is a rare soft tissue malignant neoplasm originating from neural crest cells [1].

EES has the same chromosomal translocation as Ewing sarcoma (ES), and it is currently considered that both should follow the same therapeutic strategy [2]. Pathognomonic translocations t(ll;22) (q24;q12) involving the Ewing Sarcoma gene (EWS) on chromosome 22 and the Friend leukemia integration 1 gene (FLI1) on chromosome 11 are implicated in the majority of cases [3,4].

The most important factor in determining survival in EES patients is the presence of metastases and the tumor volume at diagnosis [5], varying from 0–25% for a five-year survival rate in patients with metastatic disease, to 40–79% for those with localized disease [6].

Currently, most patients with primary localized EES may be treated using a multimodal approach. The combination of local therapy, which consists of surgery and/or radiotherapy, and adjuvant systemic chemotherapy results in recovery for approximately 75% of patients with localized tumors [7].

EES rarely involves the pelvic retroperitoneum, and in such cases, the prognosis seems to be very poor [8]. The present manuscript describes the case of a 15-year-old girl who was admitted to our department with radiological diagnostic exams showing a pelvic mass of suspected ovarian origin. In the differential diagnosis of a pelvic neoformation in a young girl, lesions of ovarian, gastrointestinal, lymph node and retroperitoneal soft tissue origin should be considered. The most common localization of the mass is the ovarian one, with a predominance of mature or immature teratoma or other neoplasms originating from the ovarian germ cells, while the most atypical localizations concern epithelial neoplasms of the ovary or retroperitoneal soft tissue neoplasms. The management of a large pelvic mass in puberty involves its removal with the most conservative surgical approach and using procedures that respect the fertility of the young patient as much as possible. The patient was therefore submitted to surgery, as indicated in the modern guidelines [9], but the surgeons found healthy genital organs and a large retroperitoneal mass of unknown origin.

## 2. Case Presentation

A 15-year-old female patient with a rapidly growing suspected pelvic mass was admitted to Fondazione Policlinico Universitario Agostino Gemelli IRCCS in April 2012 symptomatic for acute pelvic pain. Her medical history was otherwise unremarkable and preliminary laboratory investigations were normal. The girl had already had menarche and was menstruating at the time of presentation, without ever having taken any hormone therapy in her life. She had never had intercourse or gynecological visits and she was extremely frightened by her symptoms but very cooperative with health professionals. Clinical examination revealed healthy external genitalia; uterus and bladder of regular size, although strongly compressed by the pelvic mass; and a wide, solid pelvic mass, fixed to the surrounding tissues. The initial computerized tomography (CT) scan and magnetic resonance imaging (MRI) revealed a pelvic mass measuring 15 × 14 × 11 cm attributable to the right ovary, which compressed the ipsilateral ureter (Figure 1).

After hospital admission, a pelvic transabdominal ultrasound (US) was performed, and it demonstrated a solid neoformation of 18 × 14 × 13 cm vascularized at the color Doppler, suggesting a possible differential diagnosis of either a granulosa cell tumor or a yolk sac tumor of suspected ovarian origin (Figure 2). It also illustrated a bilateral dilatation of the renal pelvis. Radical surgery was therefore planned.

Intraoperative findings revealed healthy genital organs and a large retroperitoneal, retrovesical mass of about 20 cm, with a hard consistency, hypomobile with respect to the surrounding planes, tenaciously adherent to the retroperitoneal blood vessels and which caudally reached the elevatori ani muscle. The tumor was found to be partially encapsulated and showed multiple areas of necrosis. It was dissected with care and difficulty from the retroperitoneal vessels without producing any vascular damage. As the frozen section reported malignancy, the patient underwent pelvic node sampling and, because of the hydronephrosis, left ureteral stent placement. The debulking was complete and no residual tumor was left in the abdomen, ensuring adequate oncological surgical radicality (Figure 3).

The definitive diagnosis was made thanks to histopathological, immunohistochemical and real-time polymerase chain reaction (RT-PCR) examinations. To confirm the diagnosis, the pathologist analyzed the sample using an RT-PCR translocation panel for detecting gene fusion transcripts specific to Ewing’s sarcoma. The search for the gene translocations EWS on RNA extracted from the sample showed the presence of the fusion product EWS/FLI1 type 1, t(11;22) [9] (Figure 4 and Figure 5).

Figure 4 and Figure 5, A diffuse proliferation of primitive small round blue cells with round nuclei and scanty cytoplasm were observed during histopathological examinations. Mitotic activity was high. Immunohistochemical analysis revealed positive staining for vimentin, CD99, FLI1, CAM 5.2 and anti–cytokeratin antibodies AE1/AE3 and negative staining for NSE, desmin, WT1, myogenin, myoglobin, MYOD1, CD56, neurofilament, chromogranin, calretinin, synaptophysin, EMA, CD10, TTF1, S100, Melan-A, HBME-45, inhibin and smooth muscle actin. According to the pathologist, the differential diagnosis, based on the cells morphology, considered the following tumors: (1)A desmoplastic small round cell tumor (DSRCT), but this was excluded because desmin and WT1 were negative.(2)Small cell ovarian carcinoma of the hypercalcemic type (OSCCHT), but this was excluded because the ovaries were normal and the tumoral cells were FLI1+ and EMA−.(3)Metastatic small cells melanoma, but this was also excluded because of the cytokeratin positivity and negativity of HBME-45 and Melan A.(4)Rhabdomyosarcoma, excludable due to negative muscle markers (desmin, myogenin, myoglobin, MYOD1 and positive anti-cytokeratin antibodies).(5)Neuroblastoma, excluded due to negative NSE, neurofilament, synaptophysin and positive anti-cytokeratin antibodies.(6)Juvenile granulosa cell tumor of the ovary, excluded due to the intraoperative findings of normal ovaries and for negative inibin and positive anti-cytokeratin antibodies.(7)Ewing sarcoma; this was the main one suspected, both for morphological criteria and for immunohistochemistry, too (positive CD99, cytokeratin, vimentin and FLI1).

To confirm the diagnosis, the pathologist analyzed the sample using an RT-PCR translocation panel for detecting gene fusion transcripts specific to Ewing’s sarcoma.

The search for the gene translocations EWS on RNA extracted from the sample showed the presence of the fusion product EWS/FLI1 type 1, t(11;22) [9].

Postoperative radiologic systemic staging exams were scheduled. No other skeletal or extraskeletal lesions were revealed on the bone scintigraphy and CT scan. 

In the seven months following surgery, the patient was submitted to six cycles of chemotherapy according to the classical plan (Euro-Ewing 99 protocol: vincristine, ifosfamide, doxorubicin and etoposide, VIDE scheme) and two additional cycles according to another chemotherapy plan (vincristine, actinomycin D, ifosfamide, VAI scheme).

As the positron emission tomography (PET) revealed the absence of disease, the girl stopped systemic therapy to complete the treatment with 25 pelvic radiotherapy sessions. 

Before performing adjuvant chemotherapy and radiotherapy, the girl was preliminarily submitted to a new, minimally invasive surgery to remove part of the ovarian tissue for cryopreservation and ovariopexy, in order to minimize radiotherapy-induced gonadal toxicity. The ovarian transposition was aimed at removing the ovaries from the radiation field and the ovarian tissue cryopreservation was to allow transplantation in the future, permitting the recovery of ovarian function in case of its loss. Moreover, to minimize the gonadal toxicity induced by chemotherapy, during the duration of the systemic treatment and for the following five years, the patient took oral hormonal contraception with chemopreventive intent [10].

Each of the diagnostic, therapeutic and fertility preservation stages were carefully explained to the girl and her family, obtaining an excellent understanding and acceptance of the proposed clinical pathways.

Currently, the girl is alive, healthy and in optimal clinical condition, with overall survival from the first surgery of more than 11 years, no detected recurrence and regular spontaneous periods.

## 3. Discussion

Pediatric gynecology represents a complex field both from the diagnostic and therapeutic point of view and in terms of communication, both with young patients and their family. When oncological disease affects girls at a very early age, it is necessary to preserve the long-term functionality of their genital system, considering the young age of the patients at diagnosis, and to implement fertility-sparing diagnostic and therapeutic strategies.

Over the past decades, research has focused on minimally invasive approaches with the aim to save gonadal function in case of toxic (local and systemic) oncological therapies and prevent damage to the ovarian reserve following surgery for pathologies of gynecological and non-gynecological origin [11]. Pediatric patients aged under 18 years who are affected by pathologies that directly or indirectly affect the genital system deserve a multidisciplinary approach to encompass every aspect of their global health and to ensure the most effective treatment. 

The great interest to report this case is due to the difficulty of making a correct presumptive diagnosis in the pre-operatory study, which is unfortunately quite common in cases of large pelvic masses, and to the successful multidisciplinary treatment given to our patient with a favorable course of disease for more than 11 years. This case shows the importance of networking and collaborating between radiologists, pathologists, gynecologists, pediatricians, oncologists and radiotherapists in the management of pelvic mass growth in a teenage girl. A multidisciplinary team is mandatory to make an early and correct diagnosis, with appropriate surgical and imaging staging and a tailored treatment to improve therapeutic outcomes.

It is important to consider a differential diagnosis in order to choose adequate radiological methods during the diagnostic work-up, so that the best staging and treatment can be performed. The MRI findings play an important role in planning the patient’s definitive local therapy. It is necessary for a young girl presented with a pelvic mass to be submitted to transabdominal or transrectal pelvic US to study the neoformation features according to the ultrasonographic criteria available [12,13]. The second-level radiological examinations are MRIs and CT scans; from a laboratory point of view, it is possible to request ovarian tumor markers including those specific for the following germ cell tumors: CA125 (epithelial), CA19-9 (mucinous), CEA (gastrointestinal), beta chain of chorionic gonadotropin (dysgerminoma), alpha-fetoprotein (immature teratoma and/or yolk sac tumor) and inhibin b (juvenile variant granulosa cell tumor) [14]. 

In patients with EES, the tumor location is also important in determining whether surgery or a primary radiation approach is to be used for local control [15]. Regarding local therapy, there are many works which show that surgery would be the best treatment in cases of localized EES to improve survival and future quality of life. Patients not presenting metastases at diagnosis and showing a stable or good response to chemotherapy, if treated with surgery, with or without radiation therapy, seem to have better local control and a significantly higher rate of five-year event-free survival than those who are only treated with local radiation therapy [16]. A study of 57 cases of primary EES showed that the cornerstone of curative treatment in EES consists of wide surgical excision, as the authors noted five-year survival rates of 73% in those who had undergone radical wide resection [17,18]. Moreover, a radical surgical treatment, performed by surgeons with experience in oncology and analysis by expert pathologists, is mandatory in order to make the right diagnosis. This is also underlined in a study of 104 patients treated at non-tertiary centers that shows a high incidence of errors in the management [19]. The Surveillance, Epidemiology, and End Results program (SEER) points out a lack of information on the extent of surgery and type of chemotherapy and radiotherapy, which needs to be considered in drawing more definitive conclusions about ES survival and prognosis [20]. Recent evidence has confirmed the importance of a multimodal approach and a multidisciplinary team to adequately treat young ES patients, defining combinations of treatment to include surgery, radiotherapy and chemotherapy [21,22]. To obtain the right definitive diagnosis and set up the correct subsequent therapeutic strategy, the work of competent pathologists is necessary [23]. Having a competent and collaborative pathologist is an essential condition for obtaining a high-quality oncological treatment, and his active presence on the tumor board is necessary to tailor the right therapy to the patient.

Another important theme in the management of a pelvic mass diagnosed in a young girl is the fertility-sparing treatment which should consider both surgical and adjuvant therapies. Fertility preservation is becoming an integral part of cancer care among women of reproductive age. Careful selection of patients is mandatory to allow both oncological safety and an acceptable fertility outcome. Young patients submitted to fertility-sparing treatments should be addressed in a referral center with an adequate multidisciplinary team with expertise in oncology and reproductive medicine, to ensure the highest possibility of cure and to preserve reproductive function as well. Fertility-sparing programs for oncofertility are based on several risk factors such as the stage, histotype, grading, biology and natural history of the tumor. It is necessary to consider every prognostic factor linked to a specific clinical condition to tailor fertility-sparing programs to these young oncological patients. When cancer risk factors have been carefully assessed and adequate counseling has been performed with the patient and her family, it is important to evaluate fertility-sparing strategies for each phase of the treatment: surgery, radiotherapy and chemotherapy. Surgical fertility-sparing strategies for the management of a pelvic mass consist of the removal of the mass while trying to save all the soft tissues contiguous to the lesion and not involved in the neoplasm, thus saving blood vessels, nerves and parts of healthy pelvic genital organs next to the tumor. Fertility-sparing strategies related to radiotherapy concern the preservation of the ovaries from pelvic irradiation by surgically removing them from the irradiation field before starting the radiotherapy cycles. Laparoscopic ovarian transposition before pelvic radiation in young women is widely described in the literature and affects girls affected not only by malignancies of genital origin but also tumors of the gastrointestinal tract and pelvic soft tissue malignancies [24]. In the present case, a toxic chemotherapy treatment was also expected, so it was very important to plan the removal of ovarian tissue for cryopreservation at the same time as the ovarian transposition [25]. This laparoscopic intervention must obviously be scheduled in a very short time from the definitive histological diagnosis and must be performed by an expert team who can minimize the operative complications and guarantee the fastest possible recovery of the patient to avoid delay in initiating adjuvant cancer therapies. Cryopreservation of ovarian tissue consists of the removal of part of an ovary and its conservation until the moment in which the patient should experience endocrine or reproductive damage following the therapies to which she has been submitted. Ovarian tissue cryopreservation has a double meaning: It can serve as an oocyte reservoir to be subjected to orthotopic transplantation and re-establish ovulatory function if it is lost following gonadotoxic chemotherapy [26]. It can also serve to re-establish the endocrine function of the ovary, guaranteeing the production of hormones even after a heterotopic transplant and preventing the patient from premature menopause due to ovarian failure. Finally, the fertility preservation strategy to be proposed during systemic cytotoxic chemotherapy at reproductive age consists of the simultaneous administration of gonadotropin-releasing hormone agonists for prevention of chemotherapy-induced ovarian failure [27,28].

It Is mandatory not to underestimate the importance of preserving fertility in young cancer patients; it can in fact provide better care for the patient and her family, who thus identify a goal other than just survival from the tumor, and it can also produce a positive psychological effect in the patient following the need to be considered as a young woman with her own quality of life and her future wishes [29].

## 4. Conclusions

EES is a very rare tumor which affects mainly young patients, and the treatment of choice is currently still under debate.

Our case supports the possibility to propose, when technically feasible, a multimodal approach: a surgical strategy, consisting of complete debulking of the retroperitoneal tumor and, after preserving fertility, followed by systemic chemotherapy to prevent the recurrence and maintain local control of the disease by radiotherapy. 

A further aspect to consider is that radiological reports are not always completely reliable, even when taking into consideration the growing accuracy of ultrasonography, CT scans and MRI, so in cases of pelvic masses, surgery remains necessary. This is not only for the removal of the mass, resolving its compressive symptoms and performing a complete debulking without leaving residual tumors, but also to allow correct diagnosis and staging.

Fertility preservation and oncofertility are the cornerstone in the management of young patients affected by gynecological and non-gynecologic malignancies who must be referred to referral centers for their oncological safety and future quality of life. 

In conclusion, we suggest considering the possibility of atypical histological diagnosis such as EES in young patients with large pelvic or abdominal masses with uncertain diagnosis, performing surgery as soon as possible, evaluating the use of RT-PCR to detect the pathognomonic translocation, implementing a complex and all-encompassing therapeutic strategy that places the patient at the center of all clinical efforts and focusing on all clinical outcomes.

## Figures and Tables

**Figure 1 healthcare-11-01373-f001:**
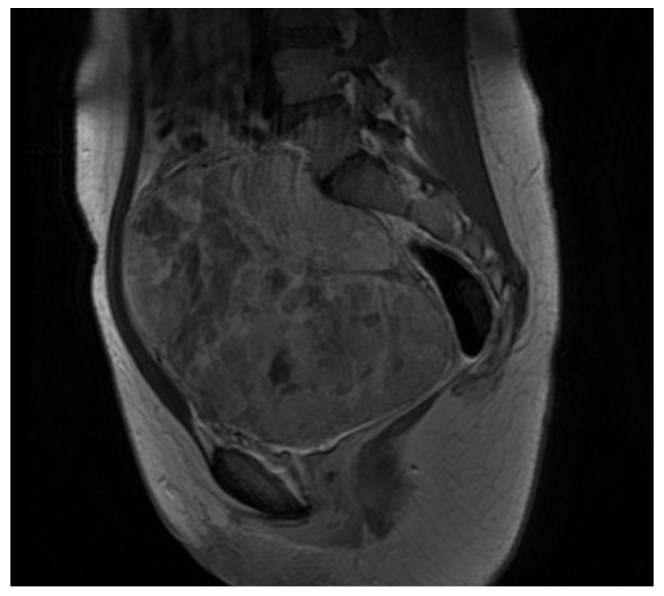
Longitudinal section, MRI.

**Figure 2 healthcare-11-01373-f002:**
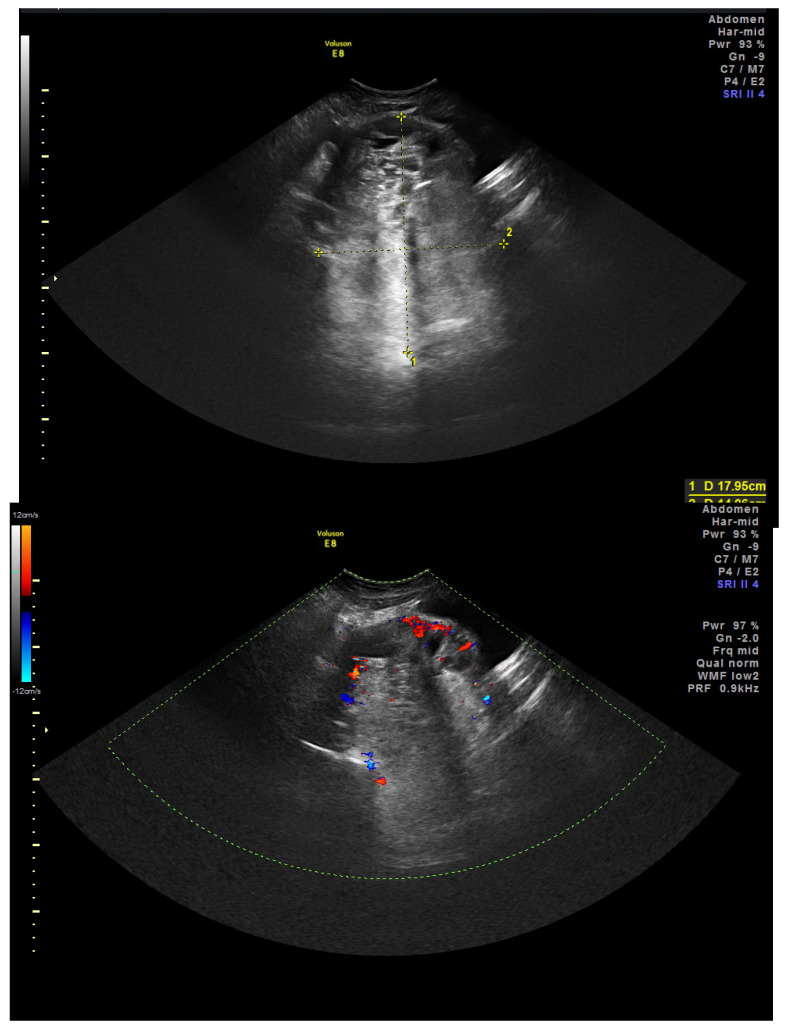
Solid pelvic mass, vascularized at the color Doppler, longitudinal diameter 17.95 cm, coronal diameter 14.06 cm, US.

**Figure 3 healthcare-11-01373-f003:**
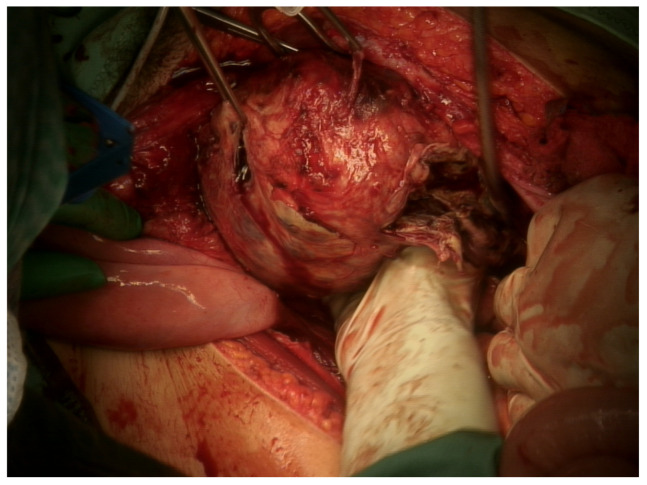
Pelvic mass, intraoperative view.

**Figure 4 healthcare-11-01373-f004:**
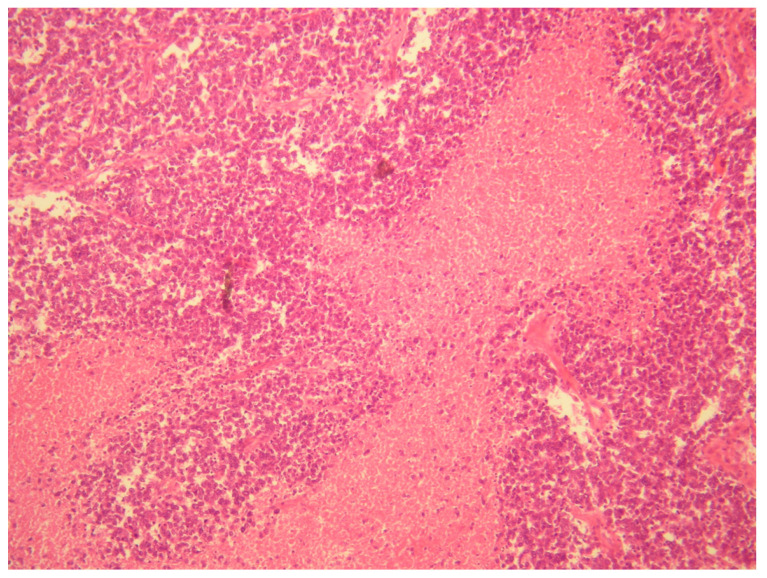
H&E stain, low enlargement (10×).

**Figure 5 healthcare-11-01373-f005:**
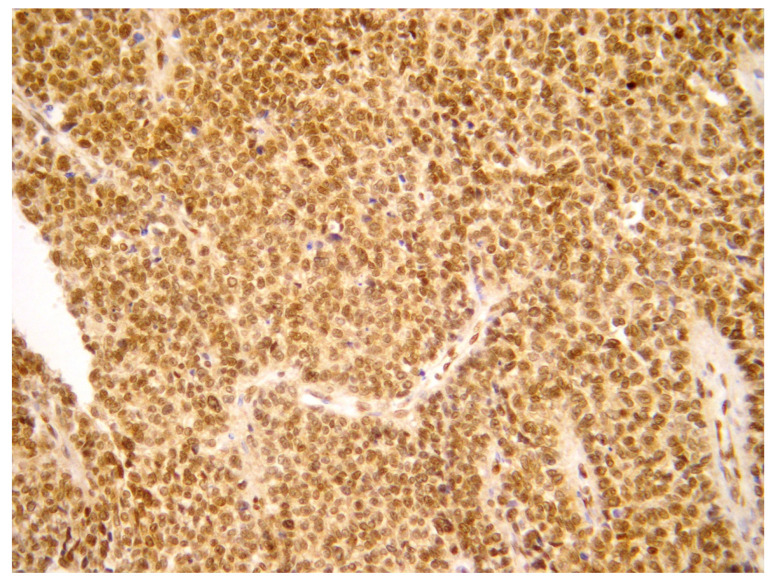
FLI1 immunohistochemistry.

## Data Availability

The data presented in this study are available on request from the corresponding author. The data are not publicly available due to privacy restrictions.

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
