# Peer review of "Large Pelvic Mass in a Female Adolescent: Atypical Presentation and Successful Treatment of Extraskeletal Ewing Sarcoma"

_healthcare, 2023, doi:10.3390/healthcare11101373_

Round 1
Reviewer 1 Report
This work aimed to show the successful multidisciplinary treatment of a pelvic mass affecting a 15-year-old female patient. The authors found the very rare definitive diagnosis of Extra skeletal Ewing Sarcoma. The work concludes that a multidisciplinary team is essential in the management of young patients affected by oncological diseases, emphasizing the role of the pathologist.
The paper is well written and conceptualized. The discussion section underline the fundamental concept of the fertility sparing program for young girls and women affected by malignancy.
Nevertheless, authors should modify the paper following some suggestion:
1) The text reported from line 89 to 117 should be included only in the figure 4,5 legend and not in the main text because this afflicts the paper fluency.
2) The authors should expand the description of the surgical findings, emphasizing the macroscopic characteristics of the detected mass and including in the discussion section a paragraph about the potential differential diagnoses more frequently associated to a pelvic mass during childhood and adolescence.
3) From line 67 to 71 the authors reported the ultrasonographic findings. I think they should include in the discussion section a paragraph about the pelvic ultrasound role in the pre-operative diagnosis of ovarian tumors including in the references the following articles: doi: 10.1136/ijgc-2021-002565; doi: 10.3389/fonc.2023.1167088.
RispondiInoltra |
Author Response
Dear Reviewer,
Thank you very much for your precious suggestions. We have proofread this manuscript according to your outstanding suggestions. Please find the revised one. Many thanks.
The text reported from line 89 to 117 should be included only in the figure 4,5 legend and not in the main text because this afflicts the paper fluency.
Reply: thank you very much for the suggestion, we will change the text to clarify it represents the figure legend.
The authors should expand the description of the surgical findings, emphasizing the macroscopic characteristics of the detected mass and including in the discussion section a paragraph about the potential differential diagnoses more frequently associated to a pelvic mass during childhood and adolescence.
Reply: thank you very much for the suggestion, we will expand the surgical findings description from line 83 to line 84 and the discussion section including more informations about the differential diagnosis for a pelvic mass during childhood.
From line 67 to 71 the authors reported the ultrasonographic findings. I think they should include in the discussion section a paragraph about the pelvic ultrasound role in the pre-operative diagnosis of ovarian tumors including in the references the following articles: doi: 10.1136/ijgc-2021-002565; doi: 10.3389/fonc.2023.1167088.
Reply: thank you very much for the suggestion, we will expand the discussion section including the role of pelvic ultrasound in diagnosis of ovarian tumors adding the references suggested.
Reviewer 2 Report
I think this study is interesting and meaningful, because the differential diagnosis of large pelvic mass like ovarian tumor may be often difficult though important. So, I have some requests as follows.
1: I want to know the figure of US with the color doppler, because you described in case presentation (page 2, line 68).
2: I want to know the figure in which the view of healthy genital organs, including uterus and ovaries, are indicated. I think that you performed the laparoscopic surgery (minimally invasive surgery) before adjuvant chemotherapy and radiotherapy (Page 5, lines 126-127). So, you can show this figure, including normal uterus and ovaries, because in this second operation, the majority of the very large pelvic mass was already removed.
Author Response
Dear Reviewer,
Thank you very much for your precious suggestions. We have proofread this manuscript according to your outstanding suggestions. Please find the revised one. Many thanks.
1: I want to know the figure of US with the color doppler, because you described in case presentation (page 2, line 68).
Reply: thank you very much for your suggestion, we will include between the figures the color doppler US of the mass as described in the text.
2: I want to know the figure in which the view of healthy genital organs, including uterus and ovaries, are indicated. I think that you performed the laparoscopic surgery (minimally invasive surgery) before adjuvant chemotherapy and radiotherapy (Page 5, lines 126-127). So, you can show this figure, including normal uterus and ovaries, because in this second operation, the majority of the very large pelvic mass was already removed.
Reply: thank you very much for your suggestion, unfortunately we didn’t performed any picture during the surgery before adjuvant radiotherapy and chemotherapy, because we genuinely thought that healthy genital organs wolud not be considered interesting to show in the present case report.
Reviewer 3 Report
The manuscript title and the abstract sound clear and clearly represent the study.
In the introduction part, please mention the most common localizations of the mass, as well as the atypical ones. Please also mention the methods used for diagnosis of pelvic mass in adolescents (could be reported in the discussion part). Suggested reference for diagnostic approach doi: 10.1080/01443615.2020.1755625.
The case is clearly presented and supported by high-quality figures. However, the text in lines 96-112 required editing as it looks and sound non-academic. If possible, please avoid long lists of bullet points.
The discussion part is interesting, however, required editing and restructuring. Too many short paragraphs-style makes it poorly comprehensible. Please rethink this part of the manuscript.
Author Response
Dear Reviewer,
Thank you very much for your precious suggestions. We have proofread this manuscript according to your outstanding suggestions. Please find the revised one. Many thanks.
In the introduction part, please mention the most common localizations of the mass, as well as the atypical ones.
Reply: thank you very much for your suggestion, we will include the most common and uncommon localizations of the mass in the introduction part from line 54 to line 59.
Please also mention the methods used for diagnosis of pelvic mass in adolescents (could be reported in the discussion part). Suggested reference for diagnostic approach doi: 10.1080/01443615.2020.1755625.
Reply: thank you very much for your suggestion, we will include the diagnosis methods used for pelvic mass in adolescents from line 212 to line 216.
The case is clearly presented and supported by high-quality figures. However, the text in lines 96-112 required editing as it looks and sound non-academic. If possible, please avoid long lists of bullet points.
Reply: thank you very much for your valuable suggestion, we will edit the text and modify it, including It in the legend of figures 4 and 5, instead that in the main text.
The discussion part is interesting, however, required editing and restructuring. Too many short paragraphs-style makes it poorly comprehensible. Please rethink this part of the manuscript.
Reply: thank you very much for your valuable suggestion, an English mother-language speaker, Prof Georgina Porro, will revise the discussion section.
Round 2
Reviewer 2 Report
Thank you for adding the figure of color doppler US. And for me personally, the figure of normal uterine adnexa, for example during laparoscopic surgeries, may be good evidence for this diagnosis. However, if these figures did not exist, I will not have further requests.
Thank you again for this chance of reviewing this research.
Author Response
Dear reviewer,thank you very much for the valuable suggestions.
We will also take advantage of your advice to consider
the differential diagnostic aspect by documenting the post-treatment laparoscopy
and we absolutely agree with you that the figure of normal uterine adnexa
may be good evidence for this diagnosis.
As you don't have further requests we thank you again for your
time and valuable review.